# Interactive Effects of Polyethylene Microplastics and Cadmium on Growth of *Microcystis aeruginosa*

**DOI:** 10.3390/toxics12040254

**Published:** 2024-03-29

**Authors:** Zihan Xue, Zetao Xiong, Zhangdong Wei, Lin Wang, Ming Xu

**Affiliations:** 1Miami College, Jinming Campus, Henan University, Kaifeng 475004, China; 17530131672@163.com (Z.X.); edwardtao26@163.com (Z.X.); 10380012@vip.henu.edu.cn (Z.W.); 2College of Geography and Environmental Science, Jinming Campus, Henan University, Kaifeng 475004, China; 3Henan Key Laboratory of Earth System Observation and Modeling, Jinming Campus, Henan University, Kaifeng 475004, China; 4Guangdong-Hong Kong Joint Laboratory for Carbon Neutrality, Jiangmen Laboratory of Carbon Science and Technology, Jiangmen 529199, China

**Keywords:** polyethylene, microplastics, cadmium, eutrophication, cyanobacteria

## Abstract

Polyethylene (PE) is a common component of microplastic pollution, and cadmium (Cd) is a prevalent pollutant in contaminated freshwater bodies in China. Among cyanobacteria, *Microcystis aeruginosa* (*M. aeruginosa*) plays a crucial role in the formation of algal blooms in these water systems. However, there has been limited research on how microplastics and heavy metals affect cyanobacteria ecologically. This study aimed to evaluate the physiological effects of individual and combined exposure to Cd pollutants and microplastics on *M. aeruginosa*. The solutions containing 13 µm and 6.5 µm PE particles (100 mg/L) with Cd were used in the research. The results indicated that the combined treatment led to a significant inhibition of chlorophyll a content, dropping to zero by day 5. The treated groups exhibited higher microcystins (MCs) content compared to the control group, suggesting increased MCs release due to pollutant exposure. Interestingly, the adsorption of heavy metals by microplastics partially alleviated the toxicity of heavy metals on algal cells. Moreover, the combined treatment significantly suppressed catalase (CAT) activity compared to Cd treatment, indicating a synergistic effect that led to greater oxidative stress. Overall, this study provides valuable insights into the impact of PE and Cd pollution on freshwater ecosystems, elucidates the physiological responses of cyanobacteria to these pollutants, and establishes a theoretical groundwork for addressing complex water pollution using cyanobacteria-based strategies.

## 1. Introduction

Microplastic pollution has emerged as a significant concern in China’s freshwater ecosystems, representing a novel and persistent pollutant [1]. Plastic debris and fibers accumulate in aquatic environments, exerting negative impacts due to their long-lasting presence [2]. Aquatic organisms, particularly phytoplankton in surface waters, are major recipients of microplastics. Studies indicate that the average abundance of microplastic particles in phytoplankton can range from 5000 to 3.42 × 10^7^ [1]. The interaction between microplastics and microalgae in aquatic environments is significant. Smaller microplastic particles can encapsulate or be internalized by algae, leading to toxicity [3] and potentially amplifying toxicity through the food web [4]. Microplastics can serve as a platform for algal cell immobilization on their surfaces, forming biofilms that alter microplastic surface chemistry and contribute to algal dispersal [5,6]. The type, particle size, and concentration of microplastics significantly influence their effects on algae. Stress on algae induced by microplastics includes processes such as membrane reconstruction, extracellular polymer (EPS) destruction, regulation of transport proteins on cell membranes, and organelle dysfunction, like mitochondrial depolarization [7,8]. Physicochemical processes in aquatic environments break down microplastics into smaller particles [9], enabling them to penetrate biological barriers and alter cell membrane permeability to enter algal cells. This effect accelerates the release of microcystins (MCs) and affects physiological processes in algal cells [10]. Nano-plastics have shown higher toxicity to *Microcystis aeruginosa* (*M. aeruginosa*) [11]. Investigations on polystyrene’s impact on algae growth revealed that smaller particle sizes (0.05–6 µm) are more toxic on that microalgae [12]. Polystyrene, polyvinyl chloride, and polypropylene inhibit the growth of *Chlorella vulgaris* within specific concentration ranges (0.01–1 g/L), with lower concentrations exhibiting greater inhibitory effects compared to higher concentrations [13,14]. Similarly, the density inhibition of methanotrophic cell populations increases as the concentration of polyvinyl chloride decreases (ranging from 0 mg/L to 0.1 mg/L) [15]. Moreover, Mao et al. [8] observed that chlorophyll a mitigates the adverse effects of polystyrene through algal-microplastic heteropolymerization during the transition from the logarithmic to stationary growth phase.

Heavy metals are commonly utilized as stabilizers and pigments in plastic production [16]. Studies have demonstrated that microplastics can serve as carriers and adsorbents of heavy metals, influencing the valence, concentration, and bioavailability of these metals [17,18]. Cyanobacteria possess mechanisms to accumulate heavy metals and mitigate their detrimental effects through intracellular binding sites, oxidation processes, and extracellular polysaccharides capable of complexing heavy metal ions [19,20]. Nonetheless, elevated concentrations of heavy metals can reduce enzyme activity in cyanobacteria, disrupt cell membrane structures, intracellular organelles like mitochondria and chloroplasts, affecting substance exchange and photosynthesis [21]. Algal cells exhibit varying levels of tolerance to different heavy metals. For instance, *M. aeruginosa* can withstand 50 mg/L Hg^2+^, while *Microcystis japonicus* shows tolerance concentrations of 0.5 mg/L Cd^2+^ and 40 mg/L Pb^2+^ [22], Iron (Fe) has been found to enhance the photosynthesis of *M. aeruginosa* [23]. The adsorption capacity of various microplastics for heavy metals varies based on the physicochemical properties of the water environment [18]. Factors like porosity and specific surface area influence the adsorption efficiency of heavy metals by microplastics, with PE notably adsorbing lead, chromium, and zinc in simulated seawater [24]. Research on phytoplankton has revealed that the presence of microplastics and Cd stress, either alone or combined, inhibits phytoplankton growth, reduces root length, and diminishes chlorophyll content [25]. Moreover, exposure of *Eichhornia crassipes* to nanomicroplastics and copper oxide (CuO) leads to root tip damage, closure of leaf stomata, and alterations in the valence state of Cu within the plant [26].

Cyanobacterial blooms, especially those induced by *M. aeruginosa*, are prevalent and harmful algal blooms [27]. However, the research on microplastics in aquatic environments has often neglected interactions with other pollutants and the potential combined stress of microplastics and other contaminants on microalgae. To address these knowledge gaps, we conducted a simulation study where we introduced polyethylene microplastics and Cd into the growth environment of *M. aeruginosa*. We evaluated parameters such as chlorophyll a, MCs, and antioxidant enzyme activity to examine the effects of polyethylene microplastics’ concentration and particle size, along with Cd, on the toxicity exerted by *M. aeruginosa*. Our study delved into how the toxicity from composite pollution manifests in the presence of *M. aeruginosa*, elucidating the underlying mechanisms governing the toxic impacts of microplastics and heavy metals on microalgae. By enhancing our understanding of the fundamental principles dictating the toxicity of microplastics and heavy metals on microalgae, we aim to comprehensively assess both the individual and combined toxic effects of microplastics and heavy metals on cyanobacteria. This research endeavor will significantly contribute to evaluating the ecological risks linked to microplastics and heavy metals in aquatic environments.

## 2. Materials and Methods

### 2.1. Experimental Design

In this study, microplastic particles were procured from Huachuang Plastic Merchants in Zhangmutou, Dongguan City, China. Two different particle sizes of polyethylene microplastics (PE-MPs), namely 13 µm and 6.5 µm, were chosen to investigate the physiological impacts of particle size on *M. aeruginosa*. The microplastics were prepared at a concentration of 100 mg/L [28]. To ensure uniform dispersion of the microplastic particles in the solution, Tween-20, constituting 1% of the total solution volume, was added. Cd, with a concentration of 0.2 mg/L, was used in the experiment and prepared by dissolving cadmium nitrate tetrahydrate [29]. The experiment was designed six treatment groups: CK (control treatment), Cd, 13 µm PE, 6.5 µm PE, 13 µm PE + Cd, and 6.5 µm PE + Cd. Three replicates were established for each of these six groups. Sampling was conducted immediately after the addition of microplastics and cadmium, followed by a cultivation period of 5 days. Subsequently, measurements of chlorophyll a, MCs, CAT, and peroxidase (POD) were carried out.

### 2.2. Algae Culture

The *M. aeruginosa* algal strain was acquired from the Shanghai Conservation Biotechnology Centre—East China Biological Resource Bank (Shanghai, China). In a sterile setting, the original algal species were introduced into a conical bottle containing BG-11 medium as detailed in Table 1. Subsequently, the conical bottle was placed in an incubator at a temperature of 25 °C, light intensity at 77.4 μmol/m^2^/s, and a light-dark cycle of 12 h each. The algal solution underwent an initial cultivation period of 6 days. About half of the original algal solution was poured into another bottle of 500 mL conical flask with BG-11 medium, and then the two bottles of algal solution were put into the incubator to continue cultivation for six days with the same conditions as before, and on the seventh day, distilled water was added to the conical flask, so that the liquid surface was level with the 500 mL scale, and at the same time 0.01 g of ammonia phosphate and 0.01 g of potassium nitrate were added to each bottle of algal solution. The two bottles of algal solution continued to cultivate in the incubator for five days, forming the experimental algal solution.

### 2.3. Determination of Chlorophyll a

The chlorophyll a content was determined in this experiment [30]. A 90% volume fraction acetone solution was prepared for sample processing. The procedure involved filtering the sample solution, using a filtration device with a glass fiber filter membrane placed at the filtration port, measuring 100 mL of sample solution from a conical flask, pouring it into the filtration device and filtrating it, and then washing the filter bib with a small amount of purified water, resulting in a green-tinted filter membrane fold at the end of filtration. Subsequently, the sample filter membrane was transferred to a mortar and pestle, where a small amount of acetone solution was added. The mixture was ground with the pestle until a paste consistency was achieved through repeated grinding. The resulting grinding solution was then transferred to a centrifuge tube, and the volume was adjusted to 10 mL with acetone solution. The centrifuge tube was placed in a light-free environment at 4 °C for immersion duration ranging from 2 h. Following immersion, centrifugation was carried out at 4000 r/min for 10 min. The supernatant in the centrifuge tube was filtered using a needle filter suction filtration method to obtain the extract, which served as the sample. Acetone solution was utilized as the comparison solution. Using a UV-3600 spectrophotometer (Shimadzu, Kumamoto, Japan), absorbance values were recorded at specific wavelengths: 664 nm (A_664_), 647 nm (A_647_), 630 nm (A_630_), and 750 nm (A_750_). The mass concentration of chlorophyll a in the sample (ρ) was determined based on the formula involving ρ_1_ (mass concentration of chlorophyll a in the specimen), *V*_1_ (constant volume of the specimen), and *V* (sampling volume)
ρ_1_ = 11.85 × (A_664_ − A_750_) − 1.54 × (A_647_ − A_750_) − 0.08 × (A_630_ − A_750_)(1)
(2)$$\rho =\frac{\rho_{1}\times V_{1}}{V}$$

### 2.4. Determination of MCs

In this experiment, the determination of MCs was conducted utilizing the algal toxin kit from Beacon Analytical Systems (Saco, ME, USA). All reagents were equilibrated to room temperature prior to use. Fifty (50) µL of enzyme markers and standards were dispensed into the experimental microtiter plate and the negative control microtiter wells of the kit using a pipette gun. Subsequently, 50 µL of antibody was added to each well of the kit microplate to homogenize the solution. The mixture was then allowed to incubate for 30 min before transferring the solution in the microtiter wells to the waste water tank swiftly. The microtiter plate was cleaned with a cleaning solution, followed by the addition of 100 µL of substrate solution and stopping, waiting 30 min and then adding 100 µL of termination solution; this step must be performed in the same order as the substrates were added. Finally, the prepared microtiter plate was placed into the enzyme marker to initiate data generation process for further analysis.

### 2.5. Enzyme Activity Measurement

#### 2.5.1. Determination of POD

In the assessment of POD, a specialized kit from Solarbio (Beijing, China) was employed. One (1) mL of POD extract was extracted from the algal solution under examination, followed by centrifugation at a speed of 4200 r/min for 30 min post-shaking and crushing. The resulting supernatant was isolated and left to equilibrate for detection at 4 °C, mirroring the protocol of our previous experimental assay [31]. Subsequent measurements were taken, with the absorbance value noted after 30 s of mixing at a wavelength of 470 nm designated as A_1_, and the absorbance value after 90 s recorded as A_2_. The difference in absorbance (∆A) was calculated as ∆A = A_2_ − A_1_. Upon completion of data acquisition, calculations were performed using the Equation (3).
POD (U/10^4^ cell) = 14.27 × ∆A(3)

#### 2.5.2. Determination of CAT

In the assessment of CAT, a specialized kit from Solarbio was employed. One (1) mL of CAT extract was introduced into the algal sap intended for testing, followed by centrifugation at 4200 r/min for 30 min post-shaking and crushing. The resulting supernatant was extracted and left to equilibrate for detection at 4 °C. The procedure remained consistent with the methodology outlined in our previous experiments [31], where the sap underwent calculations as per Equation (4).
CAT (U/10^4^ cell) = 1.356 × ∆A(4)

### 2.6. Data Processing

The data was compiled into Excel 2019 (Microsoft, Redmond, WA, USA) and presented as mean ± SD (standard deviation; *n* = 3) for analysis of statistical significance utilizing ANOVA and multiple comparisons (LSD) through SPSS 27. Statistical differences were considered highly significant when *p* < 0.01 and significant when *p* < 0.05. Subsequently, the data from all parallel groups were averaged and visualized in bar charts using Origin 2021. Throughout this study, data was presented as mean standard deviation derived from three replicates.

## 3. Results and Discussion

### 3.1. Effect of PE and Cd on Chlorophyll a Content

The importance of chlorophyll a in freshwater environments is evident through keyword co-occurrence analysis [32]. For *M. aeruginosa*, a prokaryotic organism with a simple structure, photosynthesis is an important life activity of alga [33], where chlorophyll a plays a vital role as the primary pigment [34]. In Figure 1, the findings revealed no significant disparity in chlorophyll a content among groups. However, after 5 days of exposure, a notable reduction in chlorophyll a content was observed across all treatment groups, showcasing a significant discrepancy between the pollution-exposed groups and control groups (*p* ≤ 0.01). Notably, there was no substantial difference detected between the singular and composite PE treatments. It became apparent that the inhibitory impact of lone Cd treatment on *M. aeruginosa*’s chlorophyll a content after 5 days was less pronounced compared to the solitary PE treatment. This difference was statistically significant compared to other groups. Prior studies suggest that a Cd concentration of 0.2 mg/L falls within the tolerance range of *M. aeruginosa* [22], whereas a 50 mg/L PE treatment inhibited marine microalgae’s photosynthesis over 96 h [35], hinting at a potential inhibition of chlorophyll a synthesis.

The study further observed a decreasing trend in chlorophyll a content during specific exposure periods, aligning with previous research indicating reductions in chlorophyll a content under various pollutant exposures. Furthermore, a comparison between different sizes of PE microplastics highlighted the differential inhibitory effects on chlorophyll a, potentially related to size-dependent light blocking mechanisms. Notably, combined treatments exhibited more pronounced inhibition on chlorophyll a content than individual PE or Cd treatments. Previous literature suggests synergistic toxicity effects between microplastics and Cd on *M. aeruginosa*, indicating complex interactions among pollutants and their impacts on algal photosynthesis [11].

### 3.2. Effect of PE and Cd on MCs Levels

*M. aeruginosa* responds to pollutant stress by producing MCs, which pose significant threats to ecosystems and drinking water quality. MCs, cyclic in nature, play crucial roles in photosynthesis, cellular homeostasis maintenance, adaptation to environmental changes through signaling, and are among the most concerning cyanobacterial toxins for human health [36]. At a molecular level, the synthesis and release of MCs are intricately linked to the transcription levels of relevant genes, the expression of transporter proteins, and the extent of cell membrane damage caused by pollutants [36,37,38]. The production of MCs heavily relies on the redox state and energy metabolism of photosynthesis in *M. aeruginosa*, with the synthesis also associated with reactive oxygen species accumulation, exhibiting an antioxidative stress effect [23]. Figure 2 reveals that upon the addition of pollutants on day 0, the extracellular MCs content in both the single PE-treated group and the co-treated group was elevated compared to the blank group. This increase was significant due to the regulation of MCs synthesis genes correlated with light conditions. Interestingly, there was no notable change in MCs content in the Cd-treated group compared to the control group. Previous studies by Feng et al. suggest that upregulation of MCs synthesizing genes, coupled with cell membrane damage induced by microplastics, enhances MCs release [38]. It can be inferred that microplastics adhering to algal cell surfaces create shading effects, diminishing light exposure, thereby escalating MCs synthesis and release into the extracellular space. Additionally, apart from extracellular MCs released from deceased algal cells, some MCs were also discharged into the aqueous environment due to altered cell membrane permeability. After 5 days of algal cell exposure, MCs content increased in treatment groups, except for the 6.5 μm PE + Cd group, without significant differences between groups on the fifth day. The prolonged interaction of pollutants with algal cells likely resulted in altered cell membrane permeability and increased toxicity-induced algal cell mortality, contributing to the heightened extracellular MCs content.

The findings from this experiment highlighted that on the fifth day, the single PE treatment group with smaller particle size (6.5 µm) exhibited lower MCs content compared to 13 µm PE. Notably, 0.5 µm PS demonstrated a more effective promotion of MCs synthesis and release from *M. aeruginosa* than 5 µm PS [39], while 1 μm PS was more damaging to *M. aeruginosa* than 100 nm PS [10]. Observing the activity of Cd independently revealed an increasing trend in MCs with prolonged exposure. In contrast, under combined treatment conditions, where PE possessed an adsorption capacity for heavy metal ions, the MCs content marginally decreased with longer exposure time in the presence of 6.5 µm PE. It was speculated that the bioavailability of heavy metal Cd to algal cells was diminished under these circumstances. Conversely, the phenomenon was absent in the case of 13 µm PE + Cd, suggesting that smaller-sized PE had enhanced adsorption capabilities for Cd, thereby reducing its bioavailability. Furthermore, Wang et al.’s study on the intracellular Cd content of *M. aeruginosa* indicated that groups treated with added microplastics exhibited lower Cd content than those treated with Cd alone, with nano-microplastics showing more pronounced effects [11]. Subsequent analyses uncovered a highly significant difference (*p* < 0.001) in MCs content between the combined treatment group and the single Cd-treated group on the fifth day. The combined treatment group displayed higher MCs content, indicating that the addition of PE augmented MCs levels in the presence of Cd. This suggests that PE has the potential to amplify the toxicity of Cd. However, no significant difference was observed between the combined treatment group and the single PE-treated group on the fifth day, implying that Cd’s impact on the toxicity of PE was negligible.

### 3.3. Oxidative Damage to Algae by PE and Cd

The accumulation of reactive oxygen species (ROS) in microalgae under external environmental stress [19] induces microalgae to utilize their own oxidative mechanisms in response to external environmental disturbances. Although MCs also play a role in response to antioxidant stress, the present experiments were carried out by determining the activities of enzymes in the antioxidant system (POD, CAT) to further unravel single and combined effects of microplastics and heavy metals on the physiology of *M. aeruginosa*. As the exposure time increased, there was a significant increase in POD activity between groups and CAT activity in the single treatment group, suggesting that the stress of pollutants induces oxidative stress and improves the ability of *M. aeruginosa* to adapt to the environment through an increase in the activity of antioxidant enzymes, which is similar to the changes in antioxidant enzyme activity induced by single microplastic treatment of *M. aeruginosa* [39]. From Figure 3 and Figure 4, on day 0 of exposure to the pollutants, the POD activity was maximum in the single Cd treatment group, which was significantly different compared to the other groups (*p* < 0.05), and after day 5 of exposure to the pollutants, the CAT activity of the single Cd treatment group was the highest, and the difference was highly significant (*p* < 0.001), based on which we speculated that Cd is more toxic to *M. aeruginosa* compared to PE in terms of antioxidant system. There was no significant difference between the POD activity of the combined treatment group and the single PE treatment group on day 0, but there was a significant difference in the CAT activity (*p* < 0.05), which suggests that the combined treatment increased the level of CAT activity compared to the single PE treatment, and that the addition of Cd in the presence of PE will increase the toxicity.

The experiments revealed that both POD and CAT activities on day 5 of a single 0.2 mg/L Cd treatment were greater than those on day 0. Meanwhile, the experimental results of Guo et al. showed that the activity of superoxide dismutase (SOD) of *Nostoc flageliforme* cell increased with the increase in the concentration of heavy metal exposure at heavy metal concentrations less than 10 mg/L [21]. Microplastics not only cause photoblocking of microalgal cells but also interfere with electron transfer, leading to the accumulation of excess electrons, which induces oxidative stress in *M. aeruginosa* cells, and the toxicity of the microplastics increases with time [38,40], under our combined treatment conditions, the CAT activity on day 5 was smaller than that on day 0 and significantly smaller than that on day 5 in the single Cd treatment group, suggesting that the addition of 100 mg/L PE caused stronger oxidative stress on CAT, and that the accumulation of excess electrons may have exceeded the limit of action of the antioxidant enzymes, resulting in a reduction of CAT activity, and also suggesting that there is a synergistic effect of PE and Cd on the oxidative stress of *M. aeruginosa*.

## 4. Conclusions

Microplastics can easily enter the living environment of living organisms due to their small size, and Cd contamination in natural water bodies has also caused significant environmental damage. The research aimed to investigate the physiological effects of single and combined exposure of *M. aeruginosa* to Cd contaminants in microplastic solutions. The study drew the following conclusions:(1)Chlorophyll a content of *M. aeruginosa* significantly decreased after 5 days of exposure to pollutants, with PE having a greater effect on chlorophyll a content compared to Cd, and the most intensive inhibition observed in the combined PE and Cd treatment group.(2)The impact of the 6.5 µm PE treatment group on MCs was greater compared to the 13 µm PE treatment group. It is suggested that adsorption of Cd by 6.5 µm PE occurred, resulting in reduced Cd bioavailability. Extracellular MCs content was notably higher than the control group on day 0 and day 5 in all treatment groups except the single Cd treatment group. PE increased algal MCs content, and the involvement of Cd did not significantly affect PE.(3)POD activity increased in all groups exposure at day 0. There was a synergistic effect of the combined PE and Cd treatment for 5 days on CAT activity stress, leading to decreased CAT activity in the combined treatment group. Oxidative damage caused by Cd was more noticeable. Differential impacts were noted based on PE particle sizes, with the 13 µm PE + Cd treatment group exhibiting the most stress on CAT.

In conclusion, the single and combined exposure to 100 mg/L PE and 0.2 mg/L Cd resulted in varying levels of damage to *M. aeruginosa*, affecting its photosynthetic effects, with 6.5 µm PE demonstrating higher toxicity than 13 µm PE. This study underscores the physiological impacts of microplastics and heavy metals on *M. aeruginosa* and provides data support for the effects of compound pollution on algae. At the same period, microscopic observation during the experiment revealed that *M. aeruginosa* showed shrinkage and deformation after exposure to the pollutants. Future research should delve deeper into the molecular-level physiological toxicity mechanisms of *M. aeruginosa* in response to microplastics and heavy metals, and make data support for solving the problem of algal bloom in water environment.

## Figures and Tables

**Figure 1 toxics-12-00254-f001:**
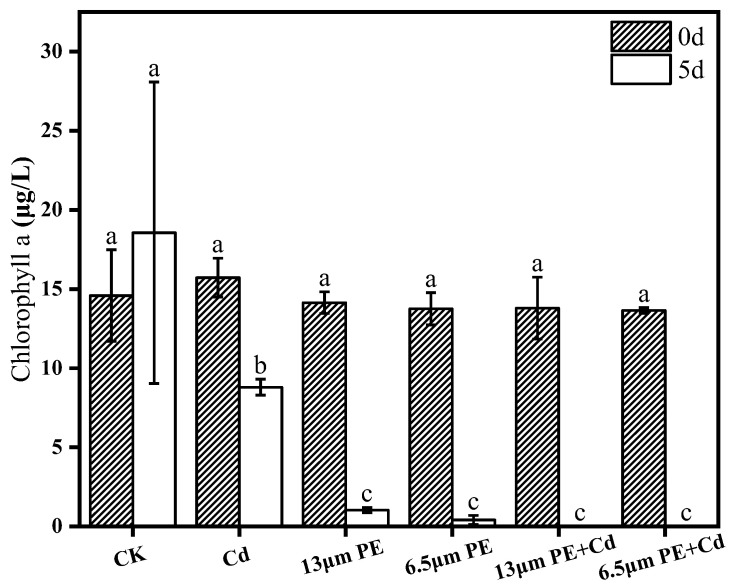
Chlorophyll a content on day 0 and day 5 of PE and Cd single and combined treatments. Different letters a, b and c indicate significant (*p* ≤ 0.05) differences between treatment groups, while the same letters indicate non-significant differences between treatments. Where: CK—Control check; Cd—Cadmium; PE—Polyethylene.

**Figure 2 toxics-12-00254-f002:**
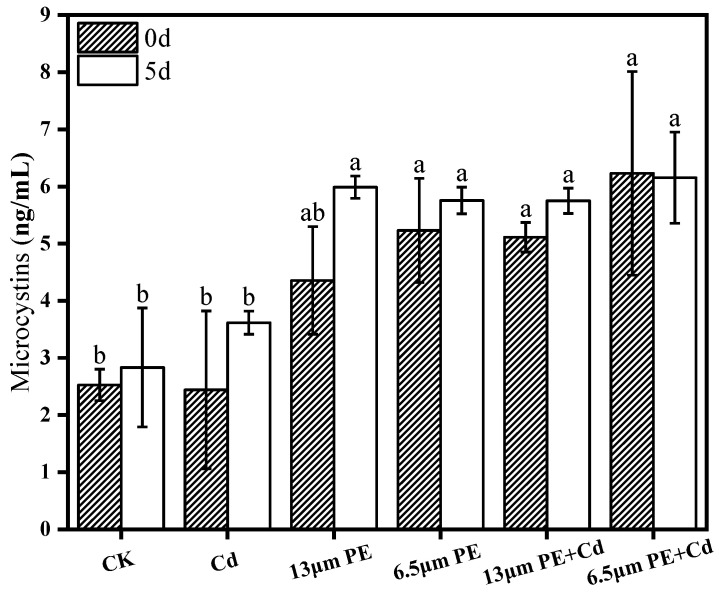
MCs content on day 0 and day 5 of PE and Cd single and combined treatments. Different letters a and b indicate significant (*p* ≤ 0.05) differences between treatment groups, while the same letters indicate non-significant differences between treatments. Where: CK—Control check; Cd—Cadmium; PE—Polyethylene.

**Figure 3 toxics-12-00254-f003:**
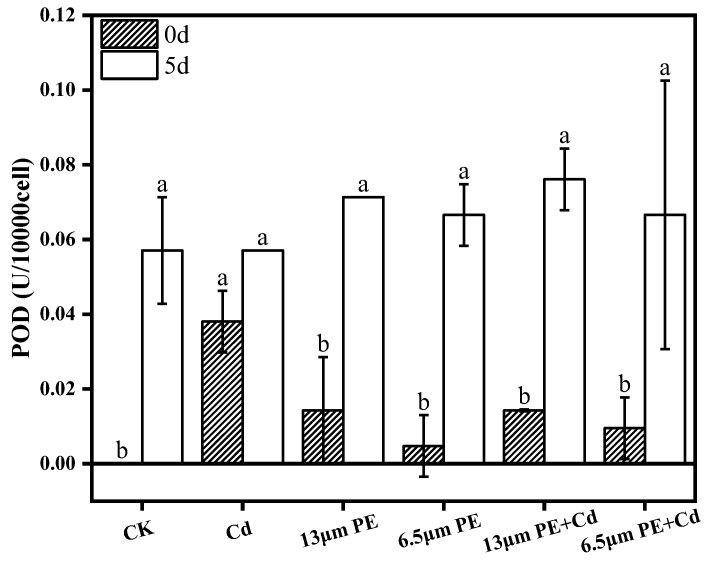
POD activity on day 0 and day 5 of PE and Cd single and combined treatments. Different letters a and b indicate significant (*p* ≤ 0.05) differences between treatment groups, while the same letters indicate non-significant differences between treatments. Where: CK—Control check; Cd—Cadmium; PE—Polyethylene.

**Figure 4 toxics-12-00254-f004:**
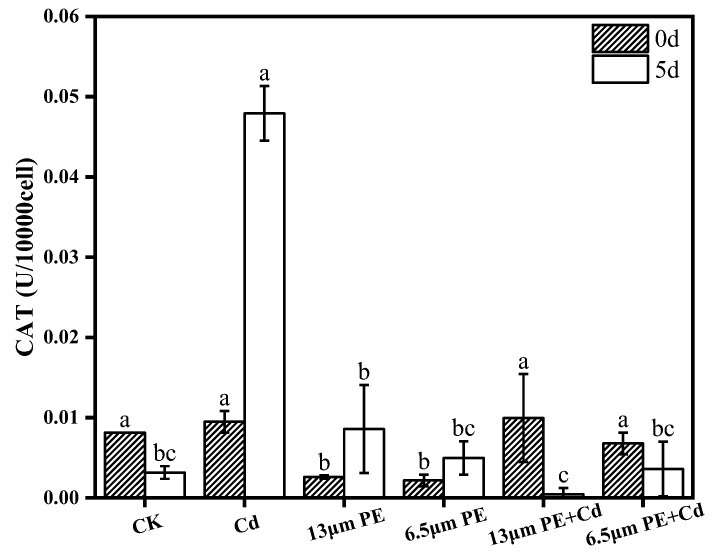
CAT activity on day 0 and day 5 of PE and Cd single and combined treatments. Different letters a, b and c indicate significant (*p* ≤ 0.05) differences between treatment groups, while the same letters indicate non-significant differences between treatments. Where: CK—Control check; Cd—Cadmium; PE—Polyethylene.

**Table 1 toxics-12-00254-t001:** BG-11 media formulations.

Ingredient	Concentration of Each Component in the Medium (mg/L)	Stock Solution (g/L)	Preparation of 1 L Culture Medium Amount of Stock Solution Required (mL)
NaNO_3_	1500	150	Mother liquor 1 10 mL
K_2_HPO_4_·3H_2_O	40	4	
Na_2_CO_3_	20	2	
CaCl_2_·2H_2_O	36	27.2	Mother liquor 2 1 mL
MgSO_4_·7H_2_O	75	7.5	Mother liquor 3 1 mL
C_6_H_8_O_7_	6	6	Mother liquor 4 1 mL
(NH_4_)_3_Fe(C_6_H_5_O_7_)_2_	6	6	
Na_2_EDTA	1	1	
H_3_BO_3_	2.86	2.86	Mother liquor 5 1 mL
MnCl_2_·4H_2_O	1.81	1.81	
ZnSO_4_·7H_2_O	0.22	0.22	
CuSO_4_·5H_2_O	0.079	0.079	
Na_2_MoO_4_·2H_2_O	0.039	0.039	

## Data Availability

The original data presented in the study are included in the article further inquiries can be directed to the corresponding author.

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
