# Peer review of "Interactive Effects of Polyethylene Microplastics and Cadmium on Growth of *Microcystis aeruginosa"

_toxics, 2024, doi:10.3390/toxics12040254_

Round 1

Reviewer 1 Report

Comments and Suggestions for Authors

Dear authors

I enjoyed reading this toxicity assessment involving Cd and microplastics and different endpoints in M. aeruginosa. Below I left some minor comments/suggestions that I hope can be useful to improve the manuscript.

1) All text: Use Microcystis aeruginosa at first appearance, then in all following appearances use M. aeruginosa.

2) Lines 40-41: The following sentence can be deleted as it is not directly related to what is being discussed in the paragraph: "In plastic-contaminated mangrove environments, several fungal species have been found capable of degrading polyethylene (PE) [6]".

3) Line 51: "... on that microalgae"

4) Lines 52-53: Chlorella vulgaris in italics. Please check the whole document for this situation, as I was able to find this repeatedly in the text.

5) Line 53: please explain that those concentrations were high and not environmentally realistic

6) Lines 69-70: Names of algae in italics.

7) Line 72: M. aeruginosa

8) Line 79: Eichhornia crassipes in italics

9) Lines 238-239: make clear that the this comparison is being with results of an experiment of other author, not the current study, since you did not test 0.5 um PS.

10) Lines 242-244: I could observed this reduction in Figure 2. Please verify.

11) Line 301: most intensive inhibition

12) Line 308: Please revise as POD activity increased only at T0, as after 5 days all treatments exhibited similar values

Author Response

Editor,

Toxics

Subject: Reviewers Responses for Manuscript “toxics-2933361”

Respected Sir,

Please find responses for the reviewer comments.

Reviewer 1

We would like to thank reviewer for reviewing our manuscript and giving their valuable suggestions. It made the regulations of our article clearer.

Scientific considerations

1) All text: Use Microcystis aeruginosa at first appearance, then in all following appearances use M. aeruginosa.1

Response: The all text has been revised accordingly to your comments.

2) Lines 40-41: The following sentence can be deleted as it is not directly related to what is being discussed in the paragraph: "In plastic-contaminated mangrove environments, several fungal species have been found capable of degrading polyethylene (PE) [6]".

Response: It has been deleted in accordance with your comments.

3) Line 51: "... on that microalgae"

Response: It has been modified in accordance with your comments.

4) Lines 52-53: Chlorella vulgaris in italics. Please check the whole document for this situation, as I was able to find this repeatedly in the text.

Response: We made the appropriate changes and identified others that needed to be changed.

5) Line 53: please explain that those concentrations were high and not environmentally realistic

Response: According to the experimental results of Wang et al. and Lin et al. the growth of Chlorella vulgaris was inhibited by 0.01g/L~1g/L microplastics, and the lower the concentration of microplastics, the greater the inhibitory effect.

6) Lines 69-70: Names of algae in italics.

Response: It has been modified in accordance with your comments.

7) Line 72: M. aeruginosa

Response: It has been modified in accordance with your comments.

8) Line 79: Eichhornia crassipes in italics

Response: It has been modified in accordance with your comments.

9) Lines 238-239: make clear that the this comparison is being with results of an experiment of other author, not the current study, since you did not test 0.5 um PS.

Response: Already changed 0.5 um PS to 13um PE.

10) Lines 242-244: I could observed this reduction in Figure 2. Please verify.

Response: A separate look at the Cd-treated group through Figure 2 revealed an increase in microcystins levels with increasing exposure time.

11) Line 301: most intensive inhibition

Response: It has been modified in accordance with your comments.

12) Line 308: Please revise as POD activity increased only at T0, as after 5 days all treatments exhibited similar values.

Response: It has been modified in accordance with your comments.

Reviewer 2 Report

Comments and Suggestions for Authors

The article contains interesting and current topics related to the interactive effects of polyethylene microplastics and cadmium on the growth of Microcystis aeruginosa. The authors presented the results of an experiment conducted on eight research groups, using various types of microplastics and cadmium, and provided a comprehensive introduction to the topic and a description of the research results. Amendments may be made to the description of materials and methods and conclusions, as well as the discussion of results, as I detail below. Once entered, the article will be suitable for publication in Toxics.
- What puzzles me most is what research sample was accepted. The repetition of all analyzes three times is commendable, but I am concerned about whether a sufficient set of data was used to properly draw conclusions. What was the sample size and why was it chosen and not another one? I would like to ask for information.
- I lack a diagram showing the individual stages of research. I would like to ask you to include it in the materials and methods.
- I would like to ask you to include in the discussion the results of the statistical analysis - descriptive statistics, scatterplots, boxplots, etc. If the sample was large enough, such analyzes should be possible to perform.
- In your conclusions, please include information on possible future research directions and the practical application of the presented results.

Author Response

Subject: Reviewers Responses for Manuscript “toxics-2933361”

Respected Sir,

Please find responses for the reviewer comments.

We would like to thank reviewer for reviewing our manuscript and giving their valuable suggestions. It made the regulations of our article clearer.

Scientific considerations

Query: What puzzles me most is what research sample was accepted. The repetition of all analyzes three times is commendable, but I am concerned about whether a sufficient set of data was used to properly draw conclusions. What was the sample size and why was it chosen and not another one? I would like to ask for information.

Response: In this study, the experiment was divided into six treatment groups simultaneously according to the differences in exposure to pollutants and each group was set up with three parallels, and chlorophyll a, microcystins and peroxidase were measured on day 0 and day 5 of pollutant addition, and the data were recorded. For the determination of the indicators, the required volume of algal solution was randomly taken from the conical flasks as the algal solution to be measured, and there were three parallels in each group, so that the sample size for each measurement was 18 samples, with a total sample size of 36 samples, and the mean and standard error of the three parallels in each group were determined, and a one-way ANOVA test was performed by comparing the mean values of each group. The reason for choosing this sample is based on the reference Wu et al. and Feng et al.'s experiments on the toxicity study of algae, their experiments chose the algal sap of algae cells after exposure to pollutants at 0h, 72h, 96h and so on as the samples for the determination of the indexes.

Query: I lack a diagram showing the individual stages of research. I would like to ask you to include it in the materials and methods.

Response: The research phase of this experiment was divided into two phases, experimental algae preparation and pollutant exposure incubation, and the determination of chlorophyll a and other indicators was carried out on day 0 and day 5 of pollutant exposure. The medium required for the experimental algae preparation was added in Materials and Methods, and the results of the indicator measurements are shown and analysed in Results and Discussion accordingly.

Query: I would like to ask you to include in the discussion the results of the statistical analysis - descriptive statistics, scatterplots, boxplots, etc. If the sample was large enough, such analyzes should be possible to perform.

Response: This part of paper combines the results and discussion in order to better demonstrate the progress and relevance of the study, this section has incorporated the results of the statistical analyses and has been supplemented accordingly based on your comments.

Query: In your conclusions, please include information on possible future research directions and the practical application of the presented results.

Response: Thank you for your comments, the conclusion contains three pieces of information about the results of this study and an outlook on future research directions.